# Zero-Cost Proxies for Lightweight NAS

**Mohamed S. Abdelfattah**[1], **Abhinav Mehrotra**[1], **Łukasz Dudziak**[1], **Nicholas D. Lane**[1,2]
[1] Samsung AI Center, Cambridge · [2] University of Cambridge
`mohamed1.a@samsung.com`

## Abstract

Neural Architecture Search (NAS) is quickly becoming the standard methodology to design neural network models. However, NAS is typically compute-intensive because multiple models need to be evaluated before choosing the best one. To reduce the computational power and time needed, a proxy task is often used for evaluating each model instead of full training. In this paper, we evaluate conventional reduced-training proxies and quantify how well they preserve ranking between neural network models during search when compared with the rankings produced by final trained accuracy. We propose a series of zero-cost proxies, based on recent pruning literature, that use just a single minibatch of training data to compute a model's score. Our zero-cost proxies use 3 orders of magnitude less computation but can match and even outperform conventional proxies. For example, Spearman's rank correlation coefficient between final validation accuracy and our best zero-cost proxy on NAS-Bench-201 is 0.82, compared to 0.61 for EcoNAS (a recently proposed reduced-training proxy). Finally, we use these zero-cost proxies to enhance existing NAS search algorithms such as random search, reinforcement learning, evolutionary search and predictor-based search. For all search methodologies and across three different NAS datasets, we are able to significantly improve sample efficiency, and thereby decrease computation, by using our zero-cost proxies. For example on NAS-Bench-101, we achieved the same accuracy $4\times$ quicker than the best previous result. Our code is made public at: `https://github.com/mohsaied/zero-cost-nas`.

## 1 Introduction

Instead of manually designing neural networks, neural architecture search (NAS) algorithms are used to automatically discover the best ones (Tan & Le, 2019a; Liu et al., 2019; Bender et al., 2018). Early work by Zoph & Le (2017) proposed using a reinforcement learning (RL) controller that constructs candidate architectures, these are evaluated and then feedback is provided to the controller based on the performance of the candidate. One major problem with this basic NAS methodology is that each evaluation is very costly – typically on the order of hours or days to train a single neural network fully. We focus on this evaluation phase – we propose using proxies that require a single minibatch of data and a single forward/backward propagation pass to score a neural network. This is inspired by recent pruning-at-initialization work by Lee et al. (2019), Wang et al. (2020) and Tanaka et al. (2020) wherein a per-parameter saliency metric is computed before training to inform parameter pruning. Can we use such saliency metrics to score an entire neural network? Furthermore, can we use these "single minibatch" metrics to rank and compare multiple neural networks for use within NAS? If so, how do we best integrate these metrics within existing NAS algorithms such as RL or evolutionary search? These are the questions that we hope to (empirically) tackle in this work with the goal of making NAS less compute-hungry. Our contributions are:

- **Zero-cost proxies** We adapt pruning-at-initialization metrics for use with NAS. This requires these metrics to operate at the granularity of an entire network rather than individual parameters – we devise and validate approaches that aggregate parameter-level metrics in a manner suitable for ranking candidates during NAS search.

- **Comparison to conventional proxies** We perform a detailed comparison between zero-cost and conventional NAS proxies that use a form of reduced-computation training. First, we quantify the rank consistency of conventional proxies on large-scale datasets: 15k models vs. 50 models used in (Zhou et al., 2020). Second, we show that zero-cost proxies can match or exceed the rank consistency of conventional proxies.

- **Ablations on NAS benchmarks** We perform ablations of our zero-cost proxies on five different NAS benchmarks (NAS-Bench-101/201/NLP/ASR and PyTorchCV) to both test the zero-cost metrics under different settings, and expose properties of successful metrics.

- **Integration with NAS** Finally, we propose two ways to use zero-cost metrics effectively within NAS algorithms: random search, reinforcement learning, aging evolution and predictor-based search. For all algorithms and three NAS datasets we show significant speedups, up to $4\times$ for NAS-Bench-101 compared to current state-of-the-art.

## 2  RELATED WORK

**NAS Efficiency**   To decrease NAS search time, various techniques were used in the literature. Pham et al. (2018) and Cai et al. (2018) use weight sharing between candidate models to decrease the training time during evaluation. Liu et al. (2019) and others use smaller datasets (CIFAR-10) as a proxy to the full task (ImageNet1k). In EcoNAS, Zhou et al. (2020) extensively investigated reduced-training proxies wherein input size, model size, number of training samples and number of epochs were reduced in the NAS evaluation phase. We compare to EcoNAS in this work to elucidate how well our zero-cost proxies perform compared to familiar and widely-used conventional proxies.

**Pruning**   The goal is to reduce the number of parameters in a neural network, one way to do this is by identifying a saliency (importance) metric for each parameter, and the less-important parameters are removed. For example, Han et al. (2015), Frankle & Carbin (2019) and others use parameter magnitudes as the criterion while LeCun et al. (1990), Hassibi & Stork (1993) and Molchanov et al. (2017) use gradients. However, the aforementioned works require training before computing the saliency criterion. A new class of pruning-at-initialization algorithms, that require no training, were introduced by Lee et al. (2019) and extended by Wang et al. (2020) and Tanaka et al. (2020). A single forward/backward propagation pass is used to compute a saliency criterion which is successfully used to heavily prune neural networks before training. We extend these pruning-at-initialization criteria towards scoring entire neural networks and we investigate their use with NAS algorithms.

**Intersection between pruning and NAS**   Concepts from pruning have been used within NAS multiple times. For example, Mei et al. (2020) use channel pruning in their AtomNAS work to arrive at customized multi-kernel-size convolutions (mixconvs as introduced by Tan & Le (2019b)). In their Blockswap work, Turner et al. (2020) use Fisher information at initialization to score different lightweight primitives that are substituted into a neural network to decrease computation. This is the earliest work we could find that attempts to perform a type of NAS by scoring neural networks without training using a pruning criterion, More recently, Mellor et al. (2020) introduced a new metric for scoring neural networks at initialization based on the correlation of Jacobians with different inputs. They perform "NAS without training" by performing random search with their zero-cost metric (`jacob_cov`) to rank neural networks instead of using accuracy. We include `jacob_cov` in our analysis and we introduce five more zero-cost metrics in this work.

## 3  PROXIES FOR NEURAL NETWORK ACCURACY

### 3.1  CONVENTIONAL NAS PROXIES (ECONAS)

In conventional sample-based NAS, a proxy training regime is often used to predict a model's accuracy instead of full training. Zhou et al. (2020) investigate conventional proxies in depth by computing the Spearman rank correlation coefficient (Spearman $\rho$) of a proxy task to final test accuracy. The proxy used is a reduced-computation training, wherein one of the following four variables is reduced: (1) number of epochs, (2) number of training samples, (3) input resolution (4) model size (controlled through the number of channels after the first convolution). Even though such proxies were used in many prior works, EcoNAS is the first systematic study of conventional proxy tasks that we found. One main finding by Zhou et al. (2020) is that using approximately $\frac{1}{4}$ of the model size and input resolution, all training samples, and $\frac{1}{10}$ the number of epochs was a reasonable proxy which yielded the best results for their experiment (Zhou et al., 2020).

### 3.2  ZERO-COST NAS PROXIES

We present alternative proxies for network accuracy that can be used to speed up NAS. A simple proxy that we use is `grad_norm` in which we sum the Euclidean norm of the gradients after a

single minibatch of training data. Other metrics listed below were previously introduced in the context of parameter pruning at the granularity of a single parameter – a saliency is computed to rank parameters and remove the least important ones. We adapt these metrics to score and rank entire neural network models for NAS.

### 3.2.1 SNIP, GRASP AND SYNAPTIC FLOW

In their `snip` work, Lee et al. (2019) proposed performing parameter pruning based on a saliency metric computed at initialization using a single minibatch of data. This saliency criteria approximates the change in loss when a specific parameter is removed. Wang et al. (2020) attempted to improve on the snip metric by approximating the change in gradient norm (instead of loss) when a parameter is pruned in their `grasp` objective. Finally, Tanaka et al. (2020) generalized these so-called *synaptic saliency* scores and proposed a modified version (`synflow`) which avoids layer collapse when performing parameter pruning. Instead of using a minibatch of training data and cross-entropy loss (as in `snip` or `grasp`), with `synflow` we compute a loss which is simply the product of all parameters in the network; therefore, no data is needed to compute this loss or the `synflow` metric itself. These are the three metrics:

$$\texttt{snip} : \mathcal{S}_p(\theta) = \left| \frac{\partial \mathcal{L}}{\partial \theta} \odot \theta \right|, \quad \texttt{grasp} : \mathcal{S}_p(\theta) = -(H \frac{\partial \mathcal{L}}{\partial \theta}) \odot \theta, \quad \texttt{synflow} : \mathcal{S}_p(\theta) = \frac{\partial \mathcal{L}}{\partial \theta} \odot \theta \tag{1}$$

where $\mathcal{L}$ is the loss function of a neural network with parameters $\theta$, $H$ is the Hessian[1], $\mathcal{S}_p$ is the per-parameter saliency and $\odot$ is the Hadamard product. We extend these saliency metrics to score an entire neural network by summing over all parameters $N$ in the model: $\mathcal{S}_n = \sum_i^N \mathcal{S}_p(\theta)_i$.

### 3.2.2 FISHER

Theis et al. (2018) perform channel pruning by removing activation channels (and their corresponding parameters) that are estimated to have the least effect on the loss. They build on the work of Molchanov et al. (2017) and Figurnov et al. (2016). More recently, Turner et al. (2020) aggregated this `fisher` metric for all channels in a convolution primitive to quantify the importance of that primitive when it is replaced by a more efficient alternative. We further aggregate the `fisher` metric for all layers in a neural network to score an entire network as shown in the following equations:

$$\texttt{fisher} : \mathcal{S}_z(z) = \left( \frac{\partial \mathcal{L}}{\partial z} z \right)^2, \quad \mathcal{S}_n = \sum_{i=1}^{M} \mathcal{S}_z(z_i) \tag{2}$$

where $\mathcal{S}_z$ is the saliency per activation $z$, and $M$ is the length of the vectorized feature map.

### 3.2.3 JACOBIAN COVARIANCE

This metric was purpose-designed to score neural networks in the context of NAS – we refer the reader to the original paper for detailed reasoning and derivation of the metric which we call `jacob_cov` (Mellor et al., 2020). In brief, this metric captures the correlation of activations within a network when subject to different inputs within a minibatch of data – the lower the correlation, the better the network is expected to perform as it can differentiate between different inputs well.

## 4 EMPIRICAL EVALUATION OF PROXY TASKS

Generally, most of the proxies presented in the previous section try to capture how *trainable* a neural network is by inspecting the gradients at the beginning of training. In this work, we refrain from attempting to explain precisely *why* each metric works (or does not work) and instead focus on the empirical evaluation of those metrics in different scenarios. We use the Spearman rank correlation coefficient (Spearman $\rho$) to quantify how well a proxy ranks models compared to the *ground-truth* ranking produced by final test accuracy (Daniel, 1990).

---

[1]The full Hessian does not need to be explicitly constructed as explained by Pearlmutter (1993).

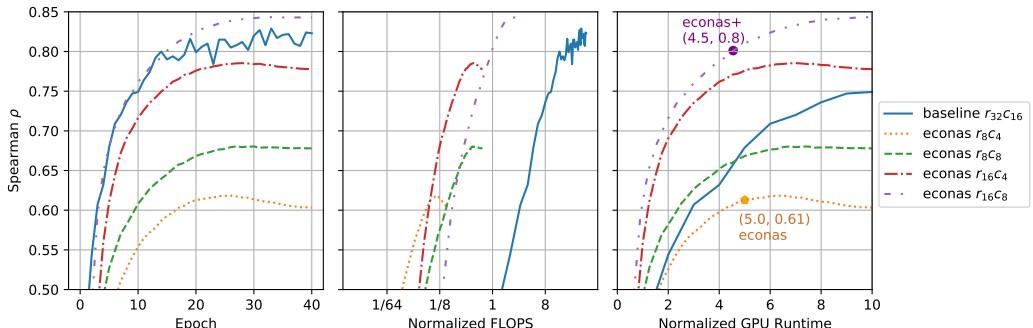

Figure 1: Evaluation of different `econas` proxies on NAS-Bench-201 CIFAR-10. FLOPS and runtime are normalized to the FLOPS/runtime of a single baseline (full training) epoch.

## 4.1 NAS-BENCH-201

NAS-Bench-201 is a purpose-built benchmark for prototyping NAS algorithms (Dong & Yang, 2020). It contains 15,625 CNN models from a cell-based search space and corresponding training statistics. We first use NAS-Bench-201 to evaluate conventional proxies from EcoNAS, then we evaluate our zero-cost proxies and compare the two approaches.

### 4.1.1 ECONAS PROXY ON NAS-BENCH-201

Even though Zhou et al. (2020) thoroughly investigated reduced-training proxies, they only evaluated a small model zoo consisting of 50 models. To study EcoNAS more extensively we evaluate it on all 15,625 models in NAS-Bench-201 search space (training details in A.1). The full configuration training of NAS-Bench-201 on CIFAR-10 uses input resolution r=32, number of channels in the stem convolution c=16 and number of epochs e=200 – we summarize this as: $r_{32}c_{16}e_{200}$.

According to the EcoNAS study, the most effective configuration divides both the input resolution and stem channels by ~4 and the number of epochs by 10, that is, $r_8c_4e_{20}$ for NAS-Bench-201 models. Keeping that in mind we investigate $r_8c_4$ in Fig. 1 (labeled `econas`); however, this proxy training seems to suffer from *overfitting* as correlation to final accuracy started to drop after 20 epochs. Additionally, the Spearman $\rho$ was a modest 0.61 when evaluated on all 15,625 models in NAS-Bench-201 – a far cry from the 0.87 achieved on the 50 models in the EcoNAS paper (Zhou et al., 2020). We additionally explore $r_8c_8$, $r_{16}c_4$ and $r_{16}c_8$ and find a very good proxy with $r_{16}c_8e_{15}$, labeled in Fig. 1 as `econas+`. From the plots in Fig. 1, we would like to highlight that:

1. A reduced-training proxy that works well on one search space may not work well on another as highlighted by the difference in Spearman $\rho$ between `econas` and `econas+`. This occurs even though both tasks in this case were CIFAR-10 image classification.

2. Even though EcoNAS-style proxies reduce computation load by a large factor (as seen in the middle plot in Fig. 1, this does not translate fully into actual runtime improvement when run on a nominal desktop GPU[2]. We therefore plot actual GPU speedup in the third subplot in Fig. 1. For example, notice that the point labeled `econas` ($r_8c_4e_{20}$) has the same FLOPS as ~$\frac{1}{10}$ of a full training epoch, but when measured on a GPU, takes time equivalent to 5 full training epochs – a 50× gap between theoretical and actual speedup.

### 4.1.2 ZERO-COST PROXIES ON NAS-BENCH-201

We now shift our focus towards our zero-cost NAS proxies which rely on gradient computations using a single minibatch of data at initialization. A clear advantage of zero-cost proxies is that they take very little time to compute – the forward/backward pass using a single minibatch of data. We ran the zero-cost proxies on all 15,625 models in NAS-Bench-201 for three image classification datasets and we summarize the results in Table 1.

The `synflow` metric performed the best on all three datasets with a Spearman $\rho$ consistently above 0.73, `jacob_cov` was second best but was also very well-correlated to final accuracy. Next came `grad_norm` and `snip` with a Spearman $\rho$ close to 0.6. We add another metric that we simply label with `vote` that takes a majority vote between the three metrics `synflow`, `jacob_cov` and

---

[2]We used Nvidia Geforce GTX 1080 Ti and ran a random sample of 10 models for 10 epochs to get an average time-per-epoch for each proxy at different batch sizes. We discuss this further in Section A.2

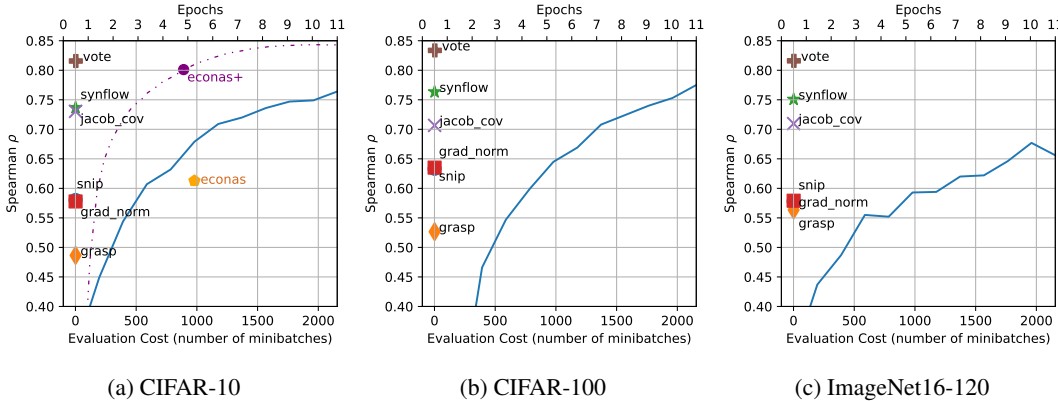

(a) CIFAR-10          (b) CIFAR-100          (c) ImageNet16-120

Figure 2: Correlation of validation accuracy to final test accuracy during the first 12 epochs of training for three datasets on the NAS-Bench-201 search space. Zero-cost and EcoNAS proxies are also labeled for comparison.

Table 1: Spearman $\rho$ of zero-cost proxies on NAS-Bench-201.

| Dataset | grad_norm | snip | grasp | fisher | synflow | jacob_cov | vote |
|---|---|---|---|---|---|---|---|
| CIFAR-10 | 0.58 | 0.58 | 0.48 | 0.36 | 0.74 | 0.73 | 0.82 |
| CIFAR-100 | 0.64 | 0.63 | 0.54 | 0.39 | 0.76 | 0.71 | 0.83 |
| ImageNet16-120 | 0.58 | 0.58 | 0.56 | 0.33 | 0.75 | 0.71 | 0.82 |

`snip` when ranking two models. This performed better than any single metric with a Spearman $\rho$ consistently above 0.8. At the cost of just 3 minibatches instead of ~1000, this is already performing slightly better than `econas+`, and much better than `econas` as shown in Fig. 2a. In Fig. 2 we also plot the rank correlation of validation accuracy (without any reduced training) over the first 10 epochs of training for the three datasets available in NAS-Bench-201.

Having set a comparison point with EcoNAS and reduced-training proxies, we have shown that zero-cost proxies can match and outperform these conventional methods in a large-scale empirical analysis. However, different NAS search spaces may behave differently, so in the remainder of this section, we test the zero-cost proxies on different search spaces.

## 4.2 MODELS IN THE WILD (PYTORCHCV)

To study zero-cost proxies in a different setting, we scored the models in the PyTorchCV database (Sémery, 2020). PytorchCV contains common state-of-the-art neural networks such as ResNets (He et al., 2016), DenseNets (Huang et al., 2017), MobileNets (Howard et al., 2017) and EfficientNets (Tan & Le, 2019a) – a representative assortment of top-performing models. We evaluated ~50 models for CIFAR-10, CIFAR-100 (Krizhevsky, 2009) and SVHN (Netzer et al., 2011), and ~200 models for ImageNet1k (Deng et al., 2009). Fig. 3 shows the resulting correlation for the zero-cost metrics. `synflow`, `snip`, `fisher` and `grad_norm` all perform similarly well on all datasets, with the exception of SVHN where `synflow` outperforms other metrics by a large margin. However, `grasp` failed in this setting completely as shown by the low mean Spearman $\rho$ and high variance as shown in Fig. 3. Curiously, `jacob_cov` also failed in this setting even though it performed well on NAS-Bench-201. This suggests that this metric is better at scoring models from within a search space (similar topology and size), but becomes worse when scoring unrelated models.

## 4.3 OTHER SEARCH SPACES

We investigate our zero-cost metrics with other NAS benchmarks. Our goal is to empirically find a good metric to speed up NAS algorithms reliably on different tasks and datasets.

- **NAS-Bench-101**: This is the first and largest NAS benchmark available with over 423k CNN models and training statistics on CIFAR-10 (Ying et al., 2019).

- **NAS-Bench-NLP**: Klyuchnikov et al. (2020) investigate the architectures of 14k different recurrent cells in natural language processing (NLP) tasks such as next word prediction.

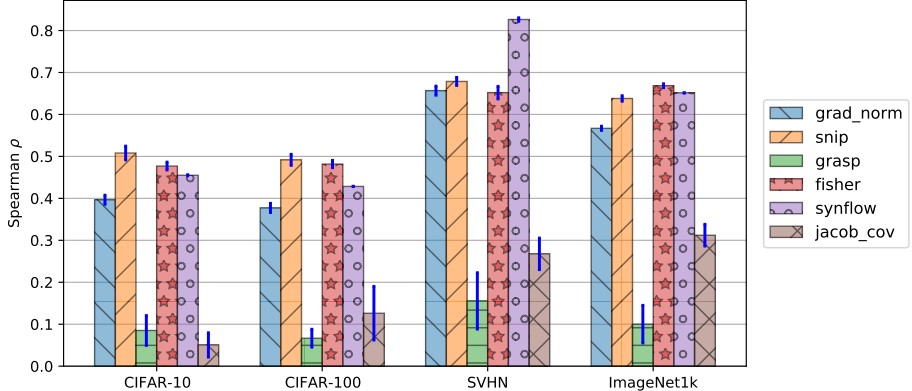

Figure 3: Performance of zero-cost metrics on PyTorchCV models (averaged over 5 seeds).

- **NAS-Bench-ASR**: This is our in-house dataset for convolution-based automatic speech recognition models evaluated on the TIMIT dataset (Garofolo et al., 1993). The search space includes linear, convolution, zeroize and skip-connections, forming 8242 models (Mehrotra et al., 2021).

Compared to NAS-Bench-201, these datasets are either much larger (NAS-Bench-101) or based on a different task (NAS-Bench-NLP/ASR). From Table 2 we would like to highlight that the `synflow` metric (highlighted in bold) is the only consistent one across all analyzed benchmarks. Additionally, even for the `synflow` metric, rank correlation is quite a bit lower than that for NAS-Bench-201 (~0.3 vs. ~0.8). Other than global rank correlation, we posit that ranking of top models from a search space is also critically important for NAS algorithms – this is because we ultimately care about finding those top models. In Section A.4 we perform an analysis of how top models are ranked by zero-cost proxies. Additionally, local rank correlation of top models could be important for NAS algorithms when two good models are compared using their proxy metric value. Tables 9 and 10 show that the only metric that maintains correct ranking among top models consistently across all NAS benchmarks is `synflow`. In Section 5 we deliberately evaluate 3 benchmarks that exhibit different levels of rank correlation: NAS-Bench-201/101/ASR to see if we can integrate `synflow` within NAS and achieve consistent gains for all three search spaces.

Table 2: Spearman $\rho$ of zero-cost proxies on other NAS search spaces.

|  | grad_norm | snip | grasp | fisher | **synflow** | jacob_cov |
|---|---|---|---|---|---|---|
| NAS-Bench-101 | 0.20 | 0.16 | 0.45 | 0.26 | **0.37** | 0.38 |
| NAS-Bench-NLP | -0.21 | -0.19 | 0.16 | – | **0.34** | 0.56 |
| NAS-Bench-ASR | 0.07 | 0.01 | – | 0.02 | **0.40** | -0.37 |

## 5 ZERO-COST NAS

Mellor et al. (2020) proposed using `jacob_cov` to score a set of randomly-sampled models and to greedily choose the model with the highest score. This "NAS without training" methodology is very attractive thanks to its simplicity and low computational cost. In this section, we evaluate our metrics in this setting that we simply call "random search" (RAND). We extend this methodology slightly: instead of just training the top model, we keep training models (from best to worst as ranked by the zero-cost metric) until the desired accuracy is achieved. However, this approach can only produce results that are as good as the metric being used – and we have no guarantees (just empirical evidence) that these metrics will perform well on all datasets. Therefore, we also investigate how to integrate zero-cost metrics within existing NAS algorithms such as reinforcement learning (RL) (Zoph & Le, 2017), aging evolution (AE) search (Real et al., 2019) and predictor-based search (Dudziak et al., 2020). More specifically, we investigate enhancing these search algorithms through either (a) zero-cost warmup phase or (b) zero-cost move proposal.

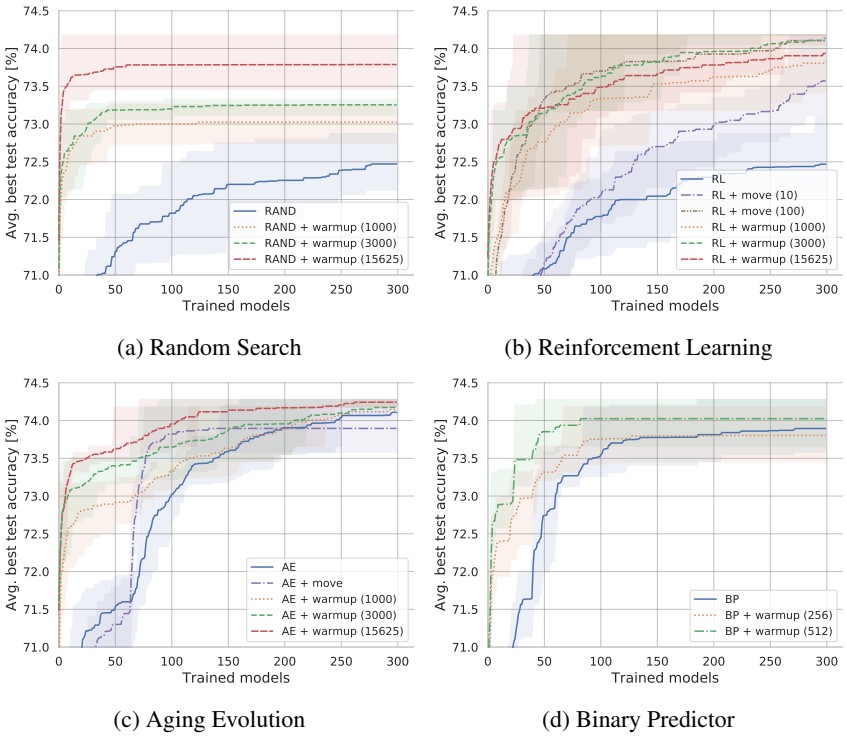

(a) Random Search

(b) Reinforcement Learning

(c) Aging Evolution

(d) Binary Predictor

Figure 4: Search speedup with the `synflow` zero-cost proxy on NAS-Bench-201 CIFAR-100.

## 5.1 ZERO-COST WARMUP

Generally speaking, by *warmup* we mean using the zero-cost proxies at the beginning of the search process to initialize the search algorithm without training any models or using accuracy. The main parameter in zero-cost warmup is the number of models for which we compute and use the zero-cost metric ($N$), and the potential gain comes from the fact that this number can be usually much larger than the number of models we can afford to train ($T \ll N$).

**Aging Evolution** We score $N$ random models with our proxy metric and choose the ones ranked highest as the initial population (pool) in the aging evolution (AE) algorithm (Real et al., 2019).

**Reinforcement Learning** In the REINFORCE algorithm (Zoph & Le (2017)), we sample $N$ random models and use their zero-cost scores to reward the controller, thus biasing it towards selecting architectures which are likely to have higher values of the chosen metrics. During warmup, reward for the controller is calculated by linearly normalizing values returned by the proxy functions to the range $[-1, 1]$ (with online adjustment of min and max).

**Binary Predictor** We warm up a binary graph convolutional network (GCN) predictor from Dudziak et al. (2020) by training it to predict relative performance of two models by considering their zero-cost scores instead of accuracy. For $N$ warmup points, we use the relative rankings (according to the zero-cost metric) of all pairs of models ($0.5N(N-1)$ pairs) when performing warmup training for the predictor. As in (Dudziak et al., 2020), models ranked by the predictor after each training round (including the warmup phase) and the top models are evaluated.

## 5.2 ZERO-COST MOVE PROPOSAL

Whereas warmup tries to leverage global correlation of the proxy metrics to the accuracy of models, move proposal focuses on a local neighborhood at each step. A common parameter for move proposal algorithms is denoted as $R$ and means sample ratio, i.e., how many models can be checked using zero-cost metrics each time we select a model to train.

**Aging Evolution** The algorithm is enhanced by performing "guided" mutations. More specifically, each time a model is being mutated (in the baseline algorithm this is done randomly) we

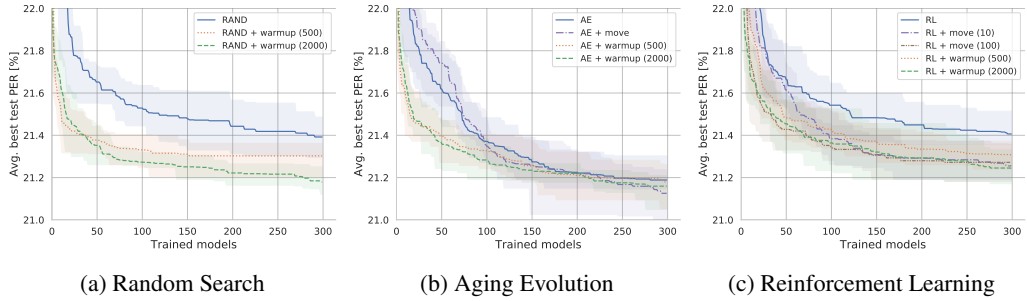

(a) Random Search          (b) Aging Evolution          (c) Reinforcement Learning

Figure 5: Search speedup with the `synflow` zero-cost proxy on NAS-Bench-ASR TIMIT.

consider all possible mutations with edit distance 1 from the current model, score them using the zero-cost proxies and select the best one to add to the pool.

**Reinforcement Learning**   In the case of REINFORCE, move proposal is similar to warmup – instead of rewarding a controller $N$ time before the search begins, we interleave $R$ zero-cost rewards for each accuracy reward ($R \ll N$).

## 5.3  RESULTS

For all NAS experiments, we repeat experiments 32 times and we plot the median and shade between the lower/upper quartiles. Our baselines are already heavily tuned and achieve the same or better results than those reported in the original NAS-Bench-101/201 papers. When adding zero-cost warmup or move proposal with `synflow`, we leave all search hyper-parameters unchanged.

**NAS-Bench-201**   The global/top-10% rank correlations of `synflow` for this dataset are (0.76/0.42) so we expect this proxy to perform quite well. Indeed, as Figure 4 and Table 7 show, we improve search speed on all four types of searches using zero-cost warmup and move proposal. RAND and RL are both significantly improved, both in terms of sample efficiency and final achieved accuracy. But even more powerful algorithms like AE and BP exhibit 5.6× and 2.3× speedups respectively to arrive at 73.5% accuracy. Generally, the more zero-cost warmup, the better the results. This holds true for all algorithms except RL which degrades at 15k warmup points, suggesting that the controller is overfitting to the `synflow` metric instead of learning to optimize for accuracy.

**NAS-Bench-101**   This dataset is an order of magnitude larger than NAS-Bench-201 and has lower global/top-10% rank correlations of (0.37/0.14). In many ways, this provides a true test as to whether these lower correlations are still useful with zero-cost warmup and move proposal. Table 3 shows a summary of the results and Figure 7 (in Section A.6) shows the full plots. As the table shows, even with modest correlations, there is a major boost to all searching algorithms thus outperforming the best previously published result by a large margin and setting a new state-of-the-art result on this dataset. However, it is worth noting that the binary predictor exhibits no improvement (but also no degradation). Perhaps this is because it was already very sample-efficient and `synflow` warmup couldn't help further due to its relatively poor correlation on this dataset.

Table 3: Comparison to prior work on NAS-Bench-101 dataset.

|  | Wen et al. (2019) | Wei et al. (2020) | Dudziak et al. (2020) | Ours | | |
|---|---|---|---|---|---|---|
|  |  |  |  | RL+M (100) | AE+W (15k) | RAND+W (3k) |
| # Trained Models | 256 | 150 | 140 | **51** | **50** | **34** |
| Test Accuracy [%] | 94.17 | 94.14 | 94.22 | **94.22** | **94.22** | **94.22** |

**NAS-Bench-ASR**   We repeat our evaluation on NAS-Bench-ASR with global/top-10% correlations (0.40/0.40). Even though this is a different task (speech recognition), `synflow` warmup and move proposal both yield large improvements in search speeds compared to all baselines in Figure 5 and Table 8. For example, to achieve a phoneme error rate (PER) of 21.3%, baseline RAND and RL required >1000 trained models, and AE required 138 trained models; however, this is reduced to 68, 173 and 87 trained models with 2000 models of zero-cost warmup.

# 6 DISCUSSION

In this section we investigate why zero-cost NAS is effective in improving the sample efficiency of NAS algorithms by looking more closely at how top models are selected by the `synflow` proxy.

**Warmup**  Table 4 shows the number of top-5% most-accurate models ranked within the top 64 models by the `synflow` metric. If we compare random warmup versus zero-cost warmup with `synflow`, random warmup will only return 5% or ~3 models out of 64 that are within the top 5% of models whereas `synflow` warmup returns a higher number of top-5% models as listed in Table 4. This is key to the improvements observed when adding zero-cost warmup to algorithms like random search or AE. For example, with AE, the numbers in Table 4 are indicative of the models that may end up in the initial AE pool. By initializing the AE pool with many good models, it becomes more likely that a random mutation will lead to an even better model, thus allowing the search to find a top model more quickly. Note that `synflow` is able to rank many good models in its top 64 models even when global/local correlation is low (as it is the case for NAS-Bench-ASR).

Table 4: Number of top-5% most-accurate models within the top 64 models returned by `synflow`.

| NAS-Bench-201 | | | NAS-Bench-101 | NAS-Bench-ASR |
|---|---|---|---|---|
| CIFAR-10 | CIFAR-100 | ImageNet16-120 | | |
| 44 | 54 | 56 | 12 | 16 |

**Move Proposal**  For a search algorithm like AE, search moves consist of random mutations (with edit distance 1 for our experiments) for a model from the AE pool. Zero-cost move proposal enhances this by trying out all possible mutations and selecting the best one according to `synflow`. To investigate how this improves search efficiency, we took 1000 random points and explored their local neighbourhood cluster of possible mutations. Table 5 shows the probability that the `synflow` proxy correctly identifies the top model. Indeed, `synflow` improves the chance of selecting the best mutation from ~4% to >30% for NAS-Bench-201 and 12% for NAS-Bench-101. Even for NAS-Bench-ASR a random mutation has a 7.7% chance ($= 1/13$) to select the best mutation, but this increases to 10% with the `synflow` proxy thus speeding up convergence to top models.

Table 5: For 1000 clusters of models with edit distance 1, we empirically measure the probability that the `synflow` proxy will select the most accurate model from each cluster.

| | NAS-Bench-201 | | | NAS-Bench-101 | NAS-Bench-ASR |
|---|---|---|---|---|---|
| | CIFAR-10 | CIFAR-100 | ImageNet16-120 | | |
| Top Model Match | 32% | 35% | 33% | 12% | 10% |
| Average Cluster Size | 25 | 25 | 25 | 26 | 13 |

# 7 CONCLUSION

In this paper, we introduced six zero-cost proxies, mainly based on recent pruning-at-initialization work, that are used to rank neural network models in NAS. First, we compared to conventional proxies (EcoNAS) that perform reduced-computation training and we found that zero-cost proxies such as `synflow` can outperform EcoNAS in maintaining rank consistency. Next, we verified our zero-cost metrics on four additional datasets of varying sizes and tasks and found that indeed out of the six initially-considered zero-cost metrics, only `synflow` was robust across all datasets for both global and top-10% rank correlation. Finally, we proposed two ways to integrate `synflow` within NAS algorithms: zero-cost warmup and zero-cost move proposal. Both methods demonstrated significant speedups across four search algorithms and three NAS benchmarks, setting new state-of-the-art results for both NAS-Bench-101 and NAS-Bench-201 datasets. Our strong and consistent empirical results suggest that the `synflow` metric, when combined with warmup and move proposal can be an effective and reliable methodology for speeding up different NAS algorithms. We hope that our work lays a foundation for further zero-cost techniques that expose favourable model properties with little computation thus making NAS more readily accessible without exorbitant computing resources. The most immediate open question for future investigation is *why* the `synflow` proxy works well – analytical insights will enable further research in zero-cost NAS proxies.

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

# A   APPENDIX

Because this paper is empirically-driven, there are many more results than what we presented in the main text of the paper. In the appendix we list many important results that support our main arguments and hypotheses in the main text of this paper.

## A.1   EXPERIMENTAL DETAILS

In Table 6 we list the hyper-parameters used in training the EcoNAS proxies to produce Figure 1. The only difference to the standard NAS-Bench-201 training pipeline (Dong & Yang, 2020) is our use of fewer epochs for the learning rate annealing schedule – we anneal the learning rate to zero over 40 epochs instead of 200. This is a common technique used in speeding up convergence for training proxies Zhou et al. (2020). We acknowledge that slightly better correlations could have been achieved for `econas` and `econas+` proxies in Figure 1 if the learning rate was annealed to zero over fewer epochs (20 and 15 epochs respectively). However, we do not anticipate the results to change significantly.

Table 6: EcoNAS training hyper-parameters for NAS-Bench-201.

| optimizer | SGD | initial LR | 0.1 |
|---|---|---|---|
| Nesterov | ✓ | final LR | 0 |
| momentum | 0.9 | LR schedule | cosine |
| weight decay | 0.0005 | epochs | 40 |
| random flip | ✓ (p=0.5) | batch size | 256 |
| random crop | ✓ | | |

One additional comment regarding Figure 1 in the main paper. While we run the training ourselves for all EcoNAS variants in the plot, we take the data for the line labeled *baseline* directly from the NAS-Bench-201 dataset. We are not sure why the line is not smooth like the lines for the EcoNAS variants that we trained but assume that this is an artifact of averaging over multiple seeds in the NAS-Bench-201 dataset. In any case, we do not anticipate that this would change any conclusions or observations that we draw from this plot.

Finally, we would like to note some details about our NAS experiments in Section 5. NAS datasets provide multiple seeds of results for each model, so whenever we "train" a model, we query a random seed from the database to mimic a real NAS pipeline without *caching*. We refer the reader to (Dudziak et al., 2020), specifically Section S3.2 for more details on this.

## A.2   GPU RUNTIME FOR ECONAS

Figure 6 shows the speedup of different EcoNAS proxies compared to baseline training. Even though $r_8 c_4$ has $64\times$ less computation compared to $r_{32} c_{16}$, it achieves a maximum of $4\times$ real speedup even when the batch size is increased.

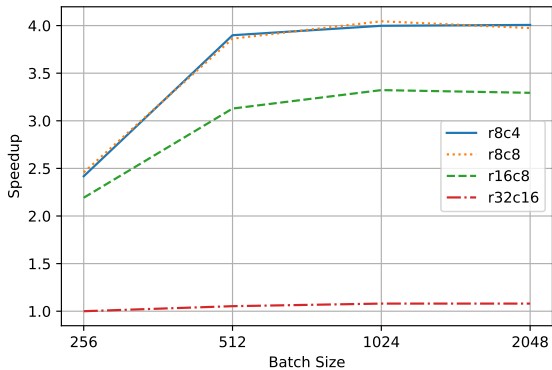

Figure 6: Higher batch sizes when training econas proxies have diminishing returns in terms of measured speedup. This measurement is done for 10 randomly-sampled NAS-Bench-201 models on the CIFAR-10 dataset.

## A.3 TABULATED RESULTS

This subsection contains tabulated results from Figures 4 and 5 to facilitate comparisons with future work. Tables 7 and 8 highlight important data points about the NAS searches we conducted with NAS-Bench-201 and NAS-Bench-ASR respectively. We highlight results in two ways: First, we show the accuracy of the best model found after 50 trained models. Second, we indicate the number of trained models needed for each search method to reach a specific accuracy (73.5% CIFAR-10 classification accuracy for NAS-Bench-201 and 21.3% TIMIT PER.) We colour the best results (red) and the second best (blue) results in each table.

Table 7: Zero-cost NAS comparison with baseline algorithms on NAS-Bench-201 CIFAR-100. We show accuracy after 50 trained models and the number of models to reach 73.5% accuracy.

| | Baseline | Warmup | | | Move | |
| | | 1000 (BP=256) | 3000 (BP=512) | 15k | 10 | 100 |
|---|---|---|---|---|---|---|
| RAND | 71.31 / 1000+ | 72.98 / 1000+ | 73.18 / 1000+ | **73.75 / 8** | – | – |
| RL | 71.08 / 1000+ | 72.76 / 145 | 73.14 / 84 | 73.21 / 107 | 71.16 / 289 | 73.34 / 70 |
| AE | 71.53 / 139 | 72.91 / 115 | 73.40 / 71 | 73.63 / 25 | | 71.3 / 77 |
| BP | 72.74 / 93 | 73.32 / 66 | **73.85 / 40** | – | – | – |

Table 8: Zero-cost NAS comparison with baseline algorithms on NAS-Bench-ASR. We show PER after 50 trained models and the number of models to reach PER=21.3%.

| | Baseline | Warmup | | Move | |
| | | 500 | 2000 | 10 | 100 |
|---|---|---|---|---|---|
| RAND | 21.65 / 1000+ | 21.38 / 1000+ | **21.35 / 68** | – | – |
| RL | 21.66 / 1000+ | 21.48 / 1000+ | 21.45 / 173 | 21.62 / 169 | 21.43 / 161 |
| AE | 21.62 / 138 | 21.40 / 115 | **21.36 / 87** | | 21.74 / 112 |

## A.4 ANALYSIS OF THE TOP 10% OF MODELS

In the main text we pointed to the fact that only `synflow` achieves consistent rank correlation for the top-10% of models across different datasets. Here, in Table 9 we provide the full results. Additionally, we hypothesized that a successful metric will rank many of the most-accurate models in its top models. In Table 10 we enumerate the percentage of top-10% most accurate models ranked as top-10% by each proxy metric. Again, `synflow` is the only consistent metric for all datasets, and performs best on average.

Table 9: Spearman $\rho$ of zero-cost proxies for the top 10% of points on all NAS search spaces.

| | grad_norm | snip | grasp | fisher | synflow | jacob_cov |
|---|---|---|---|---|---|---|
| NB2-CIFAR-10 | -0.38 | -0.38 | -0.37 | -0.38 | 0.18 | 0.17 |
| NB2-CIFAR-100 | -0.09 | -0.09 | -0.11 | -0.16 | 0.42 | 0.08 |
| NB2-ImageNet16-120 | 0.13 | 0.13 | 0.10 | 0.02 | 0.55 | 0.05 |
| NAS-Bench-101 | 0.05 | -0.01 | -0.01 | 0.07 | 0.14 | 0.08 |
| NAS-Bench-NLP | -0.03 | -0.02 | 0.04 | – | 0.10 | 0.04 |
| NAS-Bench-ASR | 0.25 | 0.13 | – | -0.07 | 0.40 | -0.03 |

## A.5 ANALYSIS OF WARMUP AND MOVE PROPOSAL

This section provides more results relevant to our discussion in Section 6. Table 11 shows the number of top-5% models ranked in the top 64 models by each metric. This is an extension to Table 4 in the main text that only shows the results for `synflow`. As shown in the table, `synflow` is the most powerful metric that we tried.

Table 12 shows the rank correlation coefficient of models within 1000 randomly-sampled local clusters of models. This result highlights that both `grad_norm` and `jacob_cov` work well in distinguishing between very similar models. However, `synflow` still consistently the best metric in this analysis. Furthermore, we measure the percentage of times that a metric correctly predicts the

Table 10: Percentage of top-10% most-accurate models within the top-10% of models ranked by each zero-cost metric.

|  | grad_norm | snip | grasp | fisher | synflow | jacob_cov |
|---|---|---|---|---|---|---|
| NB2-CIFAR-10 | 30% | 31% | 30% | 5% | 46% | 25% |
| NB2-CIFAR-100 | 35% | 36% | 34% | 4% | 50% | 24% |
| NB2-ImageNet16-120 | 31% | 31% | 32% | 5% | 44% | 30% |
| NAS-Bench-101 | 2% | 3% | 26% | 3% | 23% | 2% |
| NAS-Bench-NLP | 10% | 10% | 4% | – | 22% | 38% |
| NAS-Bench-ASR | 0% | 0% | – | 0% | 15% | 46% |

top model within a local cluster of models in Table 13 This is an extension to Table 5 in the main text. The results are averaged over 1000 randomly-sampled local clusters. Again, synflow has the highest probability of selecting the top model compared to other zero-cost metrics.

Table 11: Number of top-5% most-accurate models within the top-64 models returned by each metric.

|  | grad_norm | snip | grasp | fisher | synflow | jacob_cov |
|---|---|---|---|---|---|---|
| NB2-CIFAR-10 | 0 | 0 | 0 | 0 | 44 | 15 |
| NB2-CIFAR-100 | 4 | 4 | 4 | 0 | 54 | 16 |
| NB2-ImageNet16-120 | 13 | 13 | 14 | 0 | 56 | 15 |
| NAS-Bench-101 | 0 | 0 | 6 | 0 | 12 | 0 |
| NAS-Bench-ASR | 1 | 0 | – | 1 | 16 | 13 |

Table 12: Rank correlation coefficient for the local neighbourhoods (edit distance = 1) of 1000 clusters in each search space.

|  | grad_norm | snip | grasp | fisher | synflow | jacob_cov |
|---|---|---|---|---|---|---|
| NB2-CIFAR-10 | 0.51 | 0.51 | 0.37 | 0.37 | 0.66 | 0.62 |
| NB2-CIFAR-100 | 0.58 | 0.58 | 0.44 | 0.41 | 0.69 | 0.61 |
| NB2-ImageNet16-120 | 0.56 | 0.57 | 0.5 | 0.4 | 0.67 | 0.61 |
| NAS-Bench-101 | 0.23 | 0.21 | 0.44 | 0.27 | 0.36 | 0.37 |
| NAS-Bench-ASR | 0.59 | 0.4 | – | 0.56 | 0.38 | 0.28 |

Table 13: For 1000 clusters of points with edit distance = 1. We count the number of times wherein the top model returned by a zero-cost metric matches the top model according to validation accuracy. This represents the probability that zero-cost move proposal will perform the best possible mutation.

|  | grad_norm | snip | grasp | fisher | synflow | jacob_cov |
|---|---|---|---|---|---|---|
| NB2-CIFAR-10 | 14.8% | 14.8% | 12.7% | 5.7% | 32.2% | 14.5% |
| NB2-CIFAR-100 | 19.1% | 18.5% | 14.2% | 6.0% | 35.4% | 13.8% |
| NB2-ImageNet16-120 | 17.5% | 18.5% | 15.7% | 5.5% | 33.4% | 16.7% |
| NAS-Bench-101 | 0.4% | 0.9% | 7.4% | 0.5% | 12.3% | 0.5% |
| NAS-Bench-ASR | 11.0% | 9.8% | – | 10.3% | 10.3% | 10.5% |

## A.6 NAS-BENCH-101 SEARCH PLOTS

Figure 7 shows the NAS search curves for all considered algorithms on NAS-Bench-101 dataset. Important points from this plot are summarized in Table 3 in the main text.

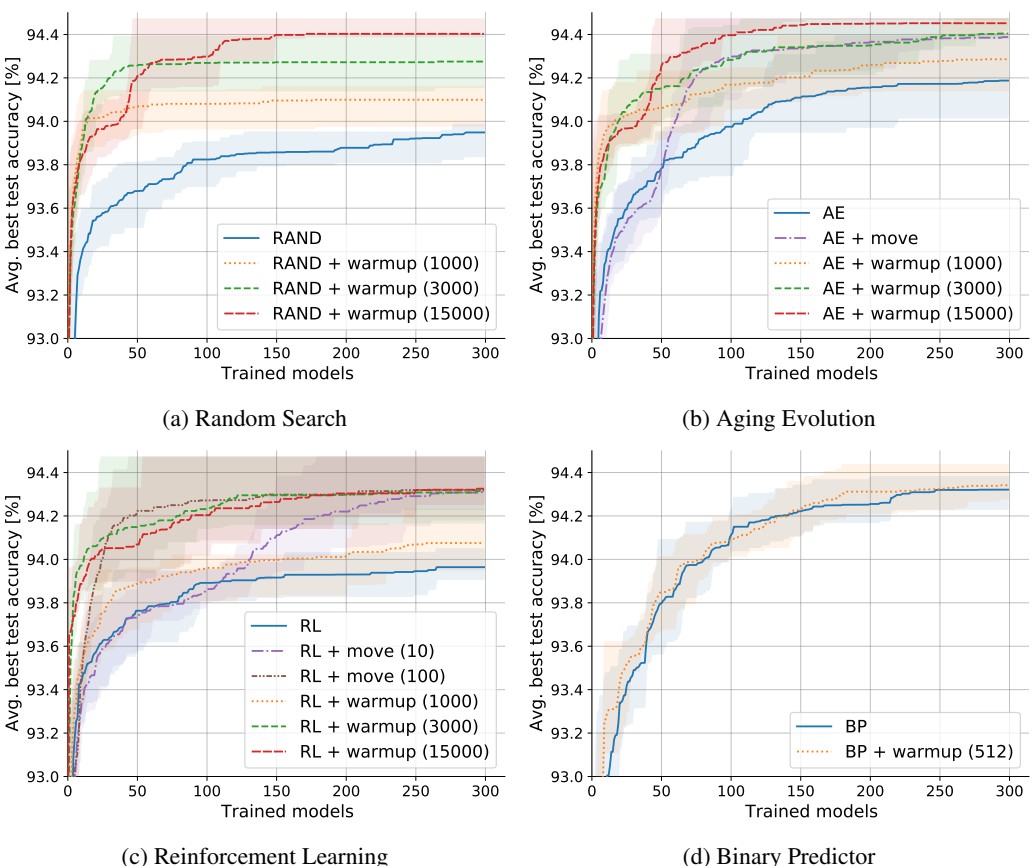

(a) Random Search

(b) Aging Evolution

(c) Reinforcement Learning

(d) Binary Predictor

Figure 7: Search speedup with the `synflow` zero-cost proxy on NAS-Bench-101 CIFAR-10.

## A.7 SENSITIVITY ANALYSIS

We performed some sensitivity analysis to investigate how the zero-cost metrics perform on all points within NAS-Bench-201 with different initialization seed, initialization method and minibatch size. We comment on each table in its caption; however, to summarize, all metrics seem to be relatively unaffected when initialization and minibatch size are varied. The one exception can be seen in Table 15 where `fisher` benefits when biases are initialized with zeroes.

Table 14: All metrics remain fairly constant when varying the initialization seed – the variations are only observed at the third significant digit. Dataload is random with 128 samples and initialization is done with default PyTorch initialization scheme.

| seed | grad_norm | snip | grasp | fisher | synflow | jacob_cov |
|---|---|---|---|---|---|---|
| 1 | 0.578 | 0.581 | 0.487 | 0.361 | 0.737 | 0.735 |
| 2 | 0.580 | 0.583 | 0.488 | 0.354 | 0.740 | 0.728 |
| 3 | 0.582 | 0.584 | 0.486 | 0.358 | 0.738 | 0.726 |
| 4 | 0.581 | 0.584 | 0.491 | 0.356 | 0.738 | 0.73 |
| 5 | 0.581 | 0.583 | 0.486 | 0.356 | 0.738 | 0.727 |
| Average | 0.580 | 0.583 | 0.488 | 0.357 | 0.738 | 0.729 |

Table 15: `fisher` becomes noticeably better when biases are initialized to zero; otherwise, metrics seem to perform independently of initialization method. Results averaged over 3 seeds.

| Weights init | Bias init | grad_norm | snip | grasp | fisher | synflow | jacob_cov |
|---|---|---|---|---|---|---|---|
| default | default | 0.580 | 0.583 | 0.488 | 0.357 | 0.738 | 0.729 |
| kaiming | default | 0.548 | 0.558 | 0.364 | 0.332 | 0.731 | 0.723 |
| xavier | default | 0.543 | 0.568 | 0.424 | 0.345 | 0.736 | 0.729 |
| default | zero | 0.581 | 0.583 | 0.488 | 0.509 | 0.738 | 0.729 |
| kaiming | zero | 0.542 | 0.551 | 0.370 | 0.479 | 0.730 | 0.723 |
| xavier | zero | 0.540 | 0.566 | 0.412 | 0.495 | 0.735 | 0.730 |

Table 16: Surprisingly, `grasp` becomes worse with more (random) data, while `grad_norm` and `snip` degrade very slightly. Other metrics seem to perform independently of the number of samples in the minibatch. Initialization is done with default PyTorch initialization scheme.

| Number of Samples | grad_norm | snip | grasp | fisher | synflow | jacob_cov |
|---|---|---|---|---|---|---|
| 32 | 0.595 | 0.596 | 0.511 | 0.362 | 0.737 | 0.732 |
| 64 | 0.589 | 0.59 | 0.509 | 0.361 | 0.737 | 0.735 |
| 128 | 0.578 | 0.581 | 0.487 | 0.361 | 0.737 | 0.735 |
| 256 | 0.564 | 0.569 | 0.447 | 0.361 | 0.737 | 0.731 |
| 512 | 0.547 | 0.552 | 0.381 | 0.361 | 0.737 | 0.724 |

## A.8 RESULTS FOR ALL ZERO-COST METRICS

Here we provide some NAS search results using all considered metrics for both RAND and AE searches on NAS-Bench-101/201 datasets. Our experiments point to `synflow` as the only effective zero-cost metric across different datasets; however, we provide the plots below for the reader to inspect how poorer metrics perform in NAS.

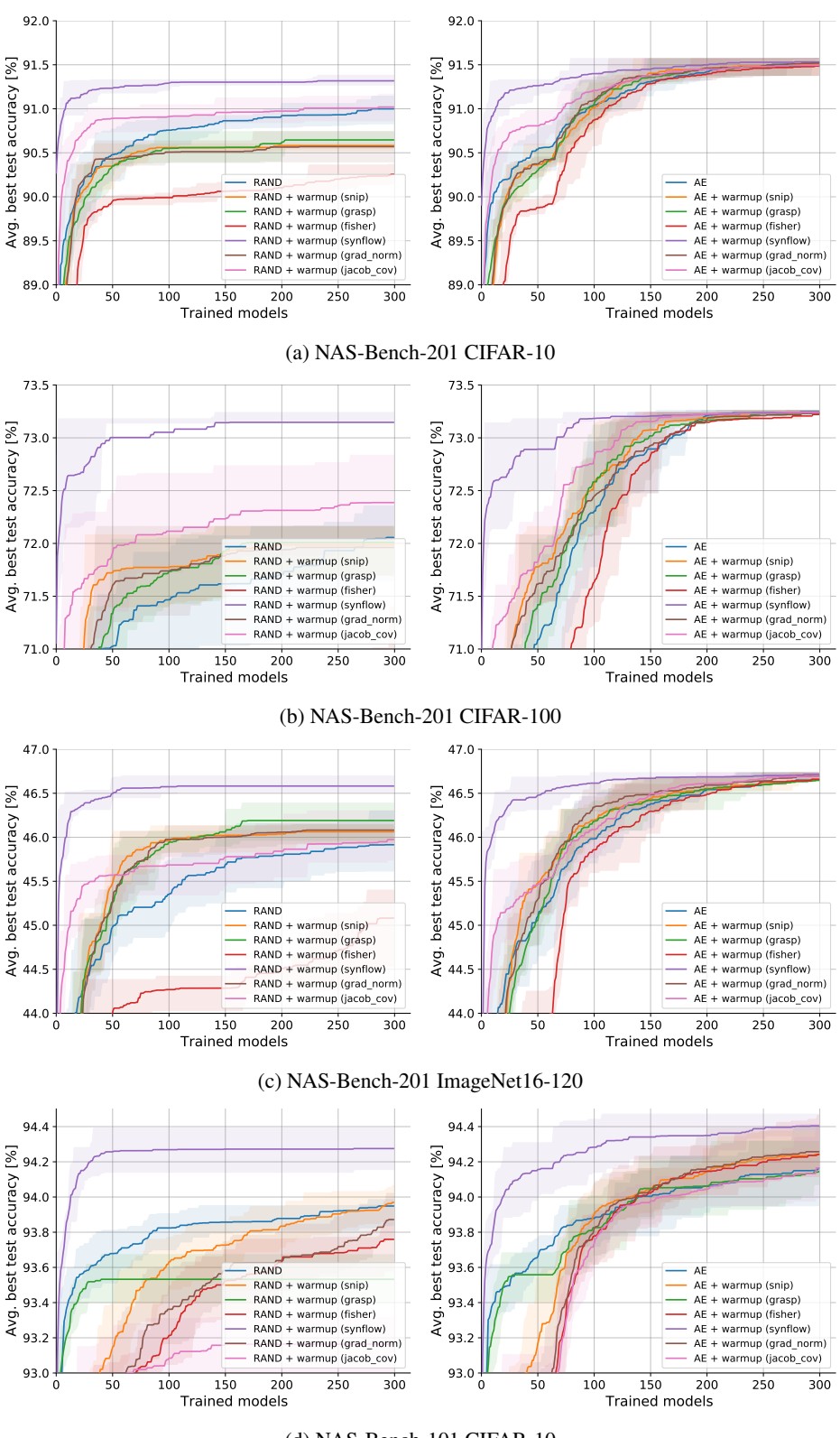

(a) NAS-Bench-201 CIFAR-10

(b) NAS-Bench-201 CIFAR-100

(c) NAS-Bench-201 ImageNet16-120

(d) NAS-Bench-101 CIFAR-10

Figure 8: Evaluation of all zero-cost proxies on different datasets and search algorithms: random search (RAND) and aging evolution (AE). RAND benefits greatly from a strong metric (such as `synflow`) but may deteriorate with a weaker metric as shown in the plot. However, AE benefits when a strong metric is used and is resilient to weaker metrics as well – it is able to recover and achieve the top accuracy in most cases.

