# OpenReview forum: "Zero-Cost Proxies for Lightweight NAS"
_ICLR.cc/2021/Conference — ICLR 2021 Poster_

### Official Review · AnonReviewer1 · 2020-10-26
**This is a paper that barely meets the acceptance criteria.**

**Rating:** 6
**Confidence:** 4

**Review:**

This paper provides an interesting direction in the neural architecture field. In particular, it proposes a series of zero-cost proxies. To design such proxies, the authors evaluate conventional reduced-training proxies and recent pruning algorithms. These proxies use just a single minibatch of training data to compute a model’s score.
Reasons for score:
Overall, I vote for accepting. On the one hand, I think the idea of utilizing pruning algorithms in NAS is very attractive since it offers a way to avoid the heavy proxy training and evaluation phase. On the other hand, the experiments in this paper are very sufficient and strong. My major concern is about the clarity of the paper. Hopefully, the authors can address my concern in the rebuttal period.

Pros:
1. The paper takes one of the most important issues of NAS: the computation burden in proxy training. For me, the problem itself is real and practical.
2. The proposed “zero-cost” proxies are very attractive for its extremely cheap computation cost. The cheap computation allows us to explore much large search space, which is of great importance in NAS. Moreover, these proxies require very little memory, which is also very attractive.
3. This paper provides extensive experiments to evaluate performances of existing pruning algorithms on a vast variety of datasets. The experimental results have a high reference value.

Cons:
1. In Eq. (1), it seems that you use a Hessian Matrix to compute grasp metric. However, the computation of a Hessian Matrix is very expensive. Is it a bug? Or you have some techniques to bypass the computation of the Hessian matrix? It would be better to provide more details here.
2. For Section 3, it would be better to provide more details about how you aggregate these metrics, which seems not very clear for me.
3. As far as I know, these techniques are not easy to integrate with some differentiable NAS algorithms, such as DARTS. However, differentiable NAS occupies an important position in all NAS methods. So, I suggest that you can point out this limitation in your paper for the sake of rigor.

Some typos：
1.  INTRODUCTION： operate at at-> operate at
2.  3.1 CONVENTIONAL NAS PROXIES (ECONAS):
 in many prior work->  in many prior works
  One main findings->  One main finding
3.  5.1 ZERO-COST WARMUP:
the number of models for which we compute -> the number of models for which we compute(T)
4.  6 CONCLUSION:
     Compute-> computing

---

> ### Author Response · Authors · 2020-11-11
> **More clarifications, references and discussion about differentiable NAS methods**
>
> We thank R1 for their detailed review and comments; we attempt to address the cons raised by the reviewer here:
> 1. Indeed, the Perlmutter trick [1] is used to compute the Hessian-vector product and in practice it is computed through automatic differentiation in the Pytorch backward pass as done in [2]. We will add text and references to the revised manuscript to point this out.
> 2. We will add more text to clarify how we aggregate the metrics. We hope that, together with equations 1 and 2, it becomes clearer for the reader. Specifically, our approach is to add the per-parameter or per-activation metrics to get a score for the whole network.
> 3. We agree that our zero-cost warmup/move algorithms are somewhat incompatible with differentiable architecture search such as DARTS. We thought of ways of integrating it with DARTS but quickly realized that speedup will not be significant. Our opinion is that DARTS is already quite fast (but has no guarantees on finding the top models). Instead, we focused on sample-based NAS algorithms, which can indeed find the best model in a search space if given enough time. We hope to make sample-based NAS algorithms (which are the slowest but most predictable) faster. We will add text in the paper to explicitly address the compatibility of our approaches with DARTS.
> 4. We thank the reviewer for kindly pointing out typos and grammar errors – all are fixed in the updated manuscript.
>
> The title of this review and the score '6' suggests that perhaps there is more that needs to be addressed - if our responses above are not sufficient, please let us know what more we can do to make our work more complete/compelling.
>
> [1] Barak A Pearlmutter. Fast exact multiplication by the Hessian. Neural computation, 6(1):147–160, 1994.
> [2] Chaoqi Wang, Guodong Zhang, and Roger Grosse. Picking winning tickets before training by preserving gradient flow. International Conference on Learning Representations (ICLR), 2020.

---

### Official Review · AnonReviewer4 · 2020-10-28
**Review of Zero-Cost Proxies for Lightweight NAS**

**Rating:** 5
**Confidence:** 4

**Review:**

############################################################

Summary

This paper provides an extensive empirical evaluation of zero-cost proxies which can be combined with existing NAS methods to speed up search time. The proposed method utilizes ‘pruning-at-initialization’ works which computes gradient-computation at initialization as a proxy for performance of the given neural architectures. Through extensive experiments, this paper compares between conventional proxies and ablation study on five NAS benchmarks and shows the validation of the proposed proxy.

############################################################

Reasons for score

Overall, I vote for weak rejecting. I like the idea of zero-cost proxies which decides the performance of neural architectures at initialization and the high utility of the zero-cost proxies available combining it with various NAS methods. However, my major concern is the limitation of zero-cost proxies where the correlations between the top 10% performance neural architectures and the values estimated by zero-cost proxies look low to use as a proxy for searching for the best neural architectures.  Hopefully, the authors can address my concern in the rebuttal period.

############################################################

Strong Points

1. This paper provides zero-cost proxies that can estimate the performance of the given neural architectures at the initialization step saving time and resources.
2. The idea of this paper is simple yet effective, where this paper is inspired by pruning-at-initialization methods and adopt those methods as proxies for NAS methods.
3. This paper demonstrates the effectiveness through extensive experiments. It first performs evaluations among several zero-cost proxies candidates to select the best one, synflow and then combines synflow and conventional NAS methods to conduct experiments on various NAS-benchmark datasets.

############################################################

Weak Points

1. My main concern is that zero-cost proxies may be not enough to use for searching the best neural architecture. The correlation between synflow which is the best one of zero-cost proxies and neural architectures for CIFAR10 from NAS201-Bench201 search space is 0.74 (Table 1). However, the correlation drops to 0.18 (Table 7 of Appendix) when we compute it between synflow, and neural architectures which have top 10% ranking performance on CIFAR10 from NAS201-Bench201 search space (Table 7). In case NAS-Bench-ASR, it drops to 0.03 (Table 7) which is nearly random. I believe while zero-cost proxies are suitable for estimate the performance tendency of all neural architectures from the search space, they are needed to be improved for the problems which search the best neural architectures. During the period, I hope the authors address my concern by performing additional experiments.
2. I think this paper has the property of the benchmark paper since this paper brings several zero-cost proxies applying it to existing NAS rather than designs a novel proxies. Thus, I expect that this paper shows more usecases combining zero-cost proxies with conventional NAS methods to demonstrate the usefulness of zero-cost proxies.
3. I recommend the results of cifar10 and imagenet120 of NAS-Bench 201 move to the original paper from the appendix.
4. Could the author shows the results of the Figure 4 as the form of the Table with exact values by selecting the results?
5. Clarity of the paper looks needed. For example,

      (1) missing reference or model name

      (2) dataset description of NAS-Bench 201 of 4.1

      (3) the detailed description of experimental setup of 4.1.2.

      (4) the detailed description of warmup technique. What meaning of (1000) of warmup (1000) in Figure 4.

---

> ### Author Response · Authors · 2020-11-11
> **More experiments to elucidate what happens with top models**
>
> We thank R4 for their questions and comments, and we attempt to address the key weaknesses that were pointed out here.
> 1. We would like to point out that our results highlight the effectiveness of zero-cost warmup and zero-cost move proposal _in spite_ of the low correlation for top models. For example, even with the low correlation of 0.14 for NAS-Bench-101 top models our zero-cost algorithms achieve up to a 4X reduction in search time, and indeed, with the top models almost uncorrelated for NAS-Bench-ASR, there is still a significant and consistent speedup as demonstrated in the paper. We agree that top-10% correlation alone does not explain the full story, so we ran more experiments to try and explain why our proxies work well given the poor rank correlation for the top-10% of models. In Table 1 shows the percentage of top-10% models predicted by a metric, that actually are in the top-10% models with the highest accuracy. Our hypothesis is that it is most important that some top models are scored highly by our metric (as shown for synflow in the table below).
>
>    **Table 1.** *Percentage of the top-10% most-accurate models that are predicted correctly to be within the top-10%.*
>
>    |                | grad_norm    | snip         | grasp        | fisher       | synflow      | jacob_cov    |
>    | :------------- | :----------: | -----------: | -----------: | -----------: | -----------: | -----------: |
>    | NB2-CIFAR-10       | 30% | 31% | 30% | 5% | **46%** | 25% |
>    | NB2-CIFAR-100      | 35% | 36% | 34% | 4% | **50%** | 24% |
>    | NB2-ImageNet16-120 | 31% | 31% | 32% | 5% | **44%** | 30% |
>    | NAS-Bench-101      | 2%  | 3%  | 26% | 3% | **23%** | 2% |
>    | NAS-Bench-ASR      | 0%  | 0%  | --  | 0% | **15%** | 44% |
>
>    * To clarify, consider the example of Aging Evolution (AE) search algorithm. This algorithm traverses the search space by exploring the neighbourhood of models within its pool, therefore, it is sufficient if zero-cost warmup selected some top models for the AE pool, and the AE algorithm will be able to quickly converge to a good result by mutating the top models within its pool. Table 2 below shows the number of top-5% models that are ranked in the top-64 models returned by each metric. As the table shows, synflow is able to place many top-5% models within the AE pool when used for warmup; therefore, this makes AE+warmup converge much faster to a good result compared to the baseline.
>
>       **Table 2**. *Number of top-5% most-accurate models within the top-64 models returned by each metric. This is representative of models that may end up in the AE pool.*
>
>       |                | grad_norm    | snip         | grasp        | fisher       | synflow      | jacob_cov    |
>       | :------------- | :----------: | -----------: | -----------: | -----------: | -----------: | -----------: |
>       | NB2-CIFAR-10       | 0 | 0 | 0 | 0 | **44** | 15 |
>       | NB2-CIFAR-100      | 4 | 4 | 4 | 0 | **54** | 16 |
>       | NB2-ImageNet16-120 | 13 | 13 | 14 | 0 | **56** | 15 |
>       | NAS-Bench-101      | 0  | 0  | 6 | 0 | **12** | 0 |
>       | NAS-Bench-ASR      | 1  | 0  | --  | 1 | **16** | 13 |
>
>    * We will attempt to add similar analyses for the other search algorithms in the paper to better explain why zero-cost NAS works well. We hope this strengthens our argument and addresses R4's main concerns.
>
> 2. We completely agree that a thorough and extensive empirical evaluation is needed for our work. We believe we achieved that by evaluating 4 NAS benchmarks and 4 competitive NAS algorithms including SOTA algorithms. Perhaps R4 can tell us which specific NAS algorithms are missing to demonstrate usefulness of our approach.
> 3. We opted to keep the results of CIFAR-10 and ImageNet120 in the appendix because they do not change any of the conclusions or trends observed with CIFAR-100. We would prefer to leave these results in the appendix due to the limited space available in the main paper – we would rather use the extra page to address R4’s main concern (in point 1). We will add text to explicitly point out that the results/trends presented in the appendix match the conclusions of the results presented in the main text on the NAS-Bench-201 benchmark.
> 4. Figure 4 contains a lot of information and tabulating everything may be infeasible; therefore, to address R4’s comment we will do two things (1) we will release all of our results alongside the codebase so others can inspect or reproduce our data where needed (2) we put a summary of Figure 4 in a table in the appendix – this may facilitate future comparisons to our work as well.
> 5. We are working hard to improve the clarity of our paper, and hope to address all R4’s clarity-related comments in the upcoming revision.

---

> > ### Comment · AnonReviewer4 · 2020-11-20
> > **Thanks for your experiments and response.**
> >
> > Thanks for your experiments and response. After carefully reading the response,   my concerns remained regarding comments 1 and 4. I  believe that this paper needs to show the concrete numbers of how much time is reduced and how much performance is increased when zero-cost is combined with the existing NAS methods to be useful for others in the NAS research community.
> >
> > [Regarding comment 1 and 4]
> >
> > Including Tables 1 and 2 in the response to comment 1, most Tables in the main paper compare between the zero-cost proxies. However, even we can choose the best proxy among them from those Tables, it is difficult to connect it to how effective the best proxy is when it is applied to existing NAS methods.
> >
> > Even this paper performed the experiments on the four benchmark datasets, except NAS-Bench-101, most results are represented as the Figures or omitted in the main paper. Also, some results for AE and BP in Figure 4 looks no effective in the final (when the performance is converged), so I feel the performance improvement is unclear.
> > As shown in Table 3 of the main paper, I believe that this paper needs to represent the results on the four benchmark sets in the form of Table 3 with concrete numbers to clearly show the effectiveness of the proposed method.
> >
> > Could the authors show the final performance and search time improvements when combining the proposed method with AE, RL, BP, and RAND on NAS-Bench-201, NAS-Bench-ASR, and NAS-Bench-NLP as Tables with concrete numbers (in the form of Table 3)? It is not a request new experiment, but to change the format of the paper's results.
> >
> > [Regarding comment 2]
> >
> > One of the main research streams for NAS is gradient-based NAS such as DARTS[1] or PC-DARTS[2]. I recommend adding the comparison with those gradient-based NAS to make the paper stronger.
> >
> > [1] DARTS: Differentiable Architecture Search, ICLR19
> >
> > [2] PC-DARTS: Partial Channel Connections for Memory-Efficient Architecture Search, ICLR2020
> >
> > I hope the authors address my concerns during the remained rebuttal period.

---

> > > ### Author Response · Authors · 2020-11-20
> > > **Adding results tables and a discussion section**
> > >
> > > Thank you for your follow-up comments and feedback.
> > >
> > > For point 1: we added Section 6 (discussion) where we expand on the investigation of warmup and move-proposal. By empirically measuring the probability of selecting top models in different search scenarios, we show that the synflow proxy is able to help NAS search converge to top models more quickly.
> > >
> > > For point 4: We followed your advice and added Tables 3 and 5 to the main text of the paper.
> > >
> > > We hope that this sufficiently addresses the points raised by R4 in their review.
> > >
> > > ----------------
> > >
> > > Regarding point 2: While our methods could be adapted to differentiable NAS methods such as DARTS, they are much more suitable for sample-based search algorithms. We decided to focus on the latter because it typically takes much more time (due to the number of models that need to be trained).

---

### Official Review · AnonReviewer3 · 2020-10-29
**Nice paper on analyzing zero-cost proxies**

**Rating:** 7
**Confidence:** 4

**Review:**

# Summary
To reduce the cost of NAS, this paper focuses on zero-cost proxies to estimate the performance of network architectures without any training.

Specifically, the authors propose a series of zero-cost proxies based on recent pruning literature. They also propose two practical strategies: zero-cost warmup and zero-cost move, to integrate zero-cost proxies into existing search algorithms (e.g., random search, RL, evolution).

Extensive experiments and analysis on NAS-Bench-101/201/NLP/ASR validate the usefulness of the proposed zero-cost proxies. Integrating zero-cost proxies into existing search algorithms significantly improves the sample efficiency.


# Strong points
1. Designing zero-cost proxies for NAS is a promising direction to reduce its cost. This paper evaluates a series zero-cost proxies and demonstrates their usefulness.
2. The idea of zero-cost warmup and zero-cost move is very appealing. Only using zero-cost proxies might not achieve competitive performance, but this idea provides a practical way of using these zero-cost proxies that will yield nice performance and a great reduction in search cost.
3. The experiments and analysis in this paper are extensive and solid, which provides many insights for the community to understand these zero-cost proxies.


# Weak points
1. “with synflow we compute a loss which is simply the product of all parameters in the network”. Why synflow using this loss? To make the paper self-contained, it will be helpful to give more clarification on this.

  Small comment: Explicitly explaining the Hadamard product notation is also helpful.

2. For some zero-cost proxies, we need to pass a minibatch of data to the network. Is the proxy sensitive or not to the choice of the minibatch? I am not sure if the “128 samples” are fixed or not in Table 4.

3. Why can r16c8 outperform the full training baseline r32c16? This seems to be counter-intuitive. Any explanation for this?

  Also, how many models are used in Figure 1, all the 15625 models in NAS-Bench-201?  It seems you will have to re-train the models under different configurations.

4. For econas+ (r16c8e15), what’s the learning rate schedule: (1) anneal to 0 (or a small number) at epoch 15, or (2) still follows the LR schedule of 200 epochs? The same question applies to other points in Figure 1. The first way (anneal to 0 at epoch 15) is more common and usually the better choice. It helps people to understand how strong this baseline is if you can confirm the implementation details in this part.

4. Figure 3 needs to be better organized to group the results for the same dataset together. It’s hard to compare different proxies in this form.

  The text says that snyflow is the best. But I notice that snip and fisher are actually slightly better than synflow, if we ignore SVHN, which is less important than ImageNet I think.

# Justification of rating
The zero-cost proxies used in this paper are mostly inspired by recent pruning literature. So, I consider this paper as more like an analysis paper that evaluates these known metrics for zero-cost proxies. But I need to point out that these known metrics were not designed for NAS initially. So, this paper provides new insights to the community.

I generally feel positive about this work due to its extensive analysis and the nice idea of zero-cost warmup/move. Most of the weak points mentioned above are for clarification. I also encourage the authors to further refine the figures and writing to make this paper better.

# Additional comments

1. Figures1&2 are blurred. The visualization style needs to be improved to make it easier to interpret.

2. Figure 6 y-axis is labeled as “normalized training time”. But from the text, the y-axis label should be speedup?

3. Table 2, the meaning of boldface is unclear.

# After rebuttal

The authors did a good job of answering my concerns. I keep my original positive rating.

---

> ### Author Response · Authors · 2020-11-11
> **Adding missing EcoNAS experimental details and text clarifications to make the paper self-contained**
>
> We thank R3 for their thorough review of our work, and we provide a response to the weak points raised here:
> 1. In our upcoming revision, we added more details on synflow (and other algorithms as well) to make the paper more self-contained instead of relying primarily on references. Additionally, we explicitly defined the symbol for the Hadamard product.
> 2. In Table 4 we vary the random minibatch (as this is also tied to the random seed). We therefore note that the choice of minibatch is not critical to establish a correlation between the zero-cost proxies and model performance. We will clarify the text in the appendix to reflect this.
> 3. All 15,625 models in NAS-Bench-201 were trained with EcoNAS proxies to produce Figure 1. For the full training case (baseline in the plot) the results were simply extracted from the NAS-Bench-201 released dataset. We believe that r16c8 outperforms full training because the learning rate annealing schedule is different. For the full training case extracted from NAS-Bench-201, the learning rate is annealed to zero over 200 epochs, whereas in the EcoNAS proxies that we ran, we anneal to zero over 40 epochs. We do this as it provides slightly faster convergence and thus makes the EcoNAS proxies stronger baselines. Ideally, we will also rerun the full training (baseline) case with the faster learning rate annealing schedule but as R3 can appreciate, this takes a lot of computation resources for the entire dataset. We will start running this experiment now but results may only be available after the rebuttal period is over.
> 4. Please see our comment above. We will add text in the manuscript to highlight the learning rate annealing schedule. Note that we annealed to 0 over 40 epochs (instead of 15 for example) because we wanted to investigate the full first 40 epochs for different EcoNAS proxies. We agree that correlation may improve slightly for EcoNAS+ if we anneal only over the first 15 epochs; however, we do not anticipate a large difference.
> 5. Thank you for the suggestion to reorganize Figure 3 – we did that in our revised paper. We agree with the reviewer that both snip and fisher are slightly better than synflow on some of the PytorchCV benchmarks – we have changed the text now to comment on this explicitly.
> 6. We appreciate the points raised in “Additional Comments”; we are working on fixing all these issues in our updated manuscript and we agree that it will significantly improve the clarity of our paper.

---

### Official Review · AnonReviewer2 · 2020-11-03
**Solid paper with extensive experiments. Results are mixed.**

**Rating:** 6
**Confidence:** 4

**Review:**

Overall, this is a solid work with interesting ingredients. The main contributions are mostly in terms of empirical results rather than a new search method.

Strengths
* The paper is very well-written and easy to follow.
* The work presented simple yet effective ways to combine zero-shot proxies with existing NAS methods: (1) to initialize standard NAS algorithms, or (2) to guide the search process. Both approaches are interesting and have well demonstrated that zero-shot proxies are complementary to existing methods.
* The paper has provided extensive evaluations over different metrics, search spaces and datasets. The results are presented in clear manner and are fairly convincing.

Weaknesses
* The technical novelty is limited in the sense that (1) the idea of zero-shot NAS is not new (Mellor et al., 2020), and (2) the best zero-shot metric (synflow) is a straightforward application of the existing work.
* Results for the rank correlation for top 10% models (A.3) are a bit disappointing because it seems to suggest zero-shot proxies are not able to identify high-performance architectures when used *alone* without leveraging other NAS methods, which are usually more expensive.
* While the method provides speedup at the early-stage, it does not seems to offer gains on the high-performance region on top of strong methods (e.g., predictive methods in Table 3).

---

> ### Author Response · Authors · 2020-11-11
> **Novelty and effectiveness of zero-cost NAS**
>
> We thank R2 for their comments and feedback, and we provide a response to the weaknesses that were pointed out:
> 1. Regarding technical novelty, we would like to point out that work by Mellor et al. is a concurrent submission to this conference (ICLR2021). Also, while we acknowledge that our zero-cost proxies are direct applications of existing zero-shot pruning work, ours is the first work to adapt these pruning metrics to the NAS domain, including enhancing existing NAS algorithms with zero-cost warmup and move proposal – given the strong and consistent results across benchmarks, we believe there is value to the NAS research community from our findings.
> 2. Indeed, correlation for top models is low, however, our results highlight the effectiveness of zero-cost warmup and zero-cost move proposal _in spite_ of the low correlation for top models. Please see [our response to R4 (point 1)](https://openreview.net/forum?id=0cmMMy8J5q&noteId=O56E1WMqM27) and the new Section 6 (Discussion) in the paper for a more detailed answer to this concern, including additional results that may help elucidate as to why zero-cost NAS is effective despite the low correlation for top models. We hope that this will sufficiently address R2's second point.
> 3. We disagree with the third weakness raised by R2. Many of the baseline NAS algorithms used in our paper are *capable* of finding the best models in our benchmarks if given enough time. Therefore, to compare NAS search methods, we look at accuracy achieved after a specific search time or the search time to achieve a specific accuracy. We were able to demonstrate that our zero-cost methods dominate all baseline NAS algorithms on either accuracy or search time.

---

### Author Response · Authors · 2020-11-18
**Overall response and revised manuscript**

The main weakness identified by R2 and R4 is the low correlation for top models exhibited by our top zero-cost proxy (synflow). In the revised manuscript  we add a new section (Section 6: Discussion), where we provided further analysis as to why we achieve large NAS gains despite the low correlation for top models.

Additionally our revised manuscript includes more edits to address other comments raised by the reviewers, including:
- Tabulating NAS-Bench-201/ASR results.
- EcoNAS experiment details. (Section A.1)
- Reworking Fig.3 to sort by dataset instead of proxy.
- Clarifications to text, notation, figures and tables as suggested by the reviewers.

---

### Decision · Program_Chairs · 2021-01-07
**Final Decision**

**Decision:**

Accept (Poster)

**Comment:**

This is a well-written paper proposing a promising a series of zero-cost proxies for NAS. Overall, the reviewers were convinced that the approach is sound and the results overall support the use of zero-cost proxies (although they are a bit weak in some cases, e.g. rank correlations in A.3). Despite some concerns amongst the reviewers around the technical novelty of the method, mostly due to the use of estimators from the pruning-at-init literature, this is promising work at the intersection of different sub-communities in ML.